# Simulation as a Tool for Understanding Experimental Observations—Ion Beam Extraction from an ECRIS

Peter Spädtke

Ingenieurbüro für Naturwissenschaft und Programmentwicklung, Junkernstr. 99, 65205 Wiesbaden, Germany;
p.spaedtke@inp-dme.de

**Abstract:** A model for the simulation of ion beam extraction from an electron-cyclotron resonance ion source is proposed. It is based on the simple fact that charged particles follow magnetic field lines. Therefore, magnetic field lines are used to generate initial conditions for ray-tracing. This model reproduces in simulation experimentally obtained results. The importance of correlations in phase-space caused by the magnetic field is shown in the simulation. This model also describes the physics of space-charge and its compensation in the extracted (fast) ion beam by low-energy electrons. Simulation provides the possibility to test theoretical assumptions, as well as to optimize technical designs.

**Keywords:** ECRIS; ion beam; extraction; ion beam transport; space-charge; simulation



## 1. Introduction

Simulation software can be used to evaluate the behavior of a physical system without performing real experiments. For this purpose, a model needs to be developed which describes the reality of such a system as precisely as possible. Incorrect or incomplete models must be avoided because they yield incorrect or incomplete results. With a good model, parameter studies can provide a reliable data space usable in standard operation of the investigated laboratory experiment so as to improve it.

The system discussed here is the formation of an ion beam from a specific kind of ion source. Various types of ion sources for different applications have been developed: Electron-Cyclotron Resonance Ion Source (ECRIS), Electron Beam Ion Source (EBIS), Laser Ion Source (LIS), MEtal Vapor Vacuum Arc (MEVVA), Philips Ionization Gauge (PIG), just to mention a few of them; more can be found in [1,2]. There will certainly be more in the future.

ECRISs were first developed in the mid 1960s by R. Geller [3] and his team. The ECRIS has the special feature of forming not only singly charged ions within the ion source plasma chamber, but also higher charge states, up to fully stripped ions. Lower m/q ratios are desirable (not only) for particle accelerators. This kind of ion source and development of the model will be described in more detail in this report. A typical design of an ECRIS is shown in Figure 1.

An ECRIS uses microwaves to create a plasma from neutral gas atoms. The plasma is confined with a suitable magnetic field. It has been found experimentally that increasing microwave power leads to not only higher charge states but also higher plasma densities. This is true for higher frequencies, too. Today, 10 GHz, 14 GHz, 18 GHz, 28 GHz, and 56 GHz klystrons and gyrotrons are available and are used worldwide in scientific laboratories, hospitals, and industrial devices. Of course, applying higher frequencies accelerates electrons to higher energies. X-rays with energies up to MeV have been measured [4] in the vicinity of an ECRIS.

With increasing frequency, the electrons are of higher energy because the resonant heating is shifted to higher energies, and stronger magnetic fields are required to keep them

within the ion-source plasma chamber. The typical magnetic field is a mirror field created by two or more coils, with a magnetic multipole field superimposed so as to improve the radial confinement; in most cases, the multipole field is a hexapole created either by permanent magnets or by a set of six coils. For lower frequencies (up to about 18 GHz) and thus also lower magnetic flux density (up to about 1 T), normal conducting coils can be used; above this limit, superconducting coils are required.

Other multipoles have been investigated also: OCTOPUS [5] is equipped with an octupole and ARC-ECRIS [6] works with a quadrupole field instead of a hexapole. These different configurations are not as easy to compare, because they influence both plasma generation and ion beam extraction and only the ion beam extraction is investigated here. Plasma generation is not simulated (yet), and certain assumptions are necessary for the plasma as, for example, plasma potential and electron temperature. The concept of temperature includes here only the low-energy plasma electrons and not the accelerated high-energy electrons which are responsible for the (step-by-step) ionization.

The ion source itself is a cylindrical vacuum chamber within a magnetic field with at least three openings in the end-plates: a gas inlet to introduce neutral gas atoms, the rf inlet to heat the electrons, and an extraction aperture to extract ions from the plasma chamber. In case of CAPRICE [7], the two openings are co-axial. The plasma chamber is biased to the extraction voltage; the ion beam energy is determined by the bias potential and the charge state.

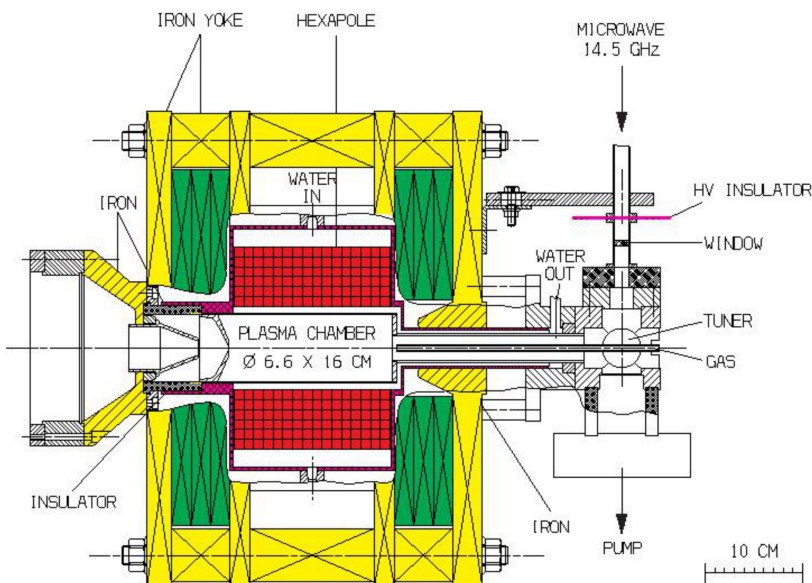

**Figure 1.** The Caprice ECRIS. Normal conducting coils (shown in green) together with a permanent magnet hexapole (shown in red). Microwave power is fed to the source from a high power generator at 14.5 GHz, connected by a wave guide to the ECRIS. The bias voltage needs to be isolated from the microwave. Iron parts in yellow. Extraction is to the left; here, a simple diode system is shown. Gas is injected from the side opposite to the extraction side.

This type of ion source has been improved over the years, accompanied by biannual international conferences (ICIS) and workshops (ECRIS). On each of these occasions, new intensity records for specific ion beams or higher charge states are usually presented. The improvements are obtained by new hardware components (higher power, higher frequency, higher magnetic flux density, etc.) or by specific tricks (mixing of frequencies, mixing of different gases or mixing of different isotopes, etc.). Computer simulation has been used to accomplish these developments, but the state of the art of the available hardware and software was limited and a correct model was not available in the beginning.

## 2. Development of the Model

Simulation of ion beam formation utilizes at least two different differential equations: a boundary value system to describe the electric scalar potential and the magnetic vector potential, and an initial value problem to solve for the ion trajectories. Only the time-independent solution is required for cw operation of the ion source:

$$\nabla \cdot \mathbf{E} = \rho / \epsilon_0 \qquad (1)$$

and

$$\nabla \cdot \mathbf{B} = 0 \qquad (2)$$

over the volume of interest. $\mathbf{E}$ is the electric field, $\mathbf{B}$ the magnetic flux density, $\rho$ the space-charge, $\epsilon_0$ the dielectric constant. $\nabla$ describes the nabla vector operator. $\partial \mathbf{E} / \partial t$ and $\partial \mathbf{B} / \partial t$ are assumed to be zero. These differential equations (Equations (1) and (2)) can be solved when boundary conditions are known.

To solve for the charged particle trajectories, the initial coordinate conditions need to be known. These initial coordinates are three spatial coordinates and three velocity coordinates. The solution of these trajectories can be found by two-fold integration and provides a space-charge distribution which influences the solution of Equation (1); an iterative procedure is used to find a self-consistent solution.

Particle-In-Cell (PIC) codes, taking the interaction of single particles into account, can be used to solve for the ion motion, but in the steady-state case, a trajectory code is sufficient.

*Adaptation to ECRIS*

A variety of different models have been developed in the past for different charged particle sources: a geometrically fixed emitting surface, as for thermionic electron sources or surface ionization ion sources, is certainly not a realistic model for an ECRIS; and the assumption of a homogeneous plasma does not show the experimentally well-known ion beam structure formed by an ECRIS.

In an ECRIS, a mirror-like B-field of the order of 1 T (and above) is combined with a radial B-field component created by a hexapole. The influence of the magnetic hexapole flux density at a typical extraction aperture radius of 3 mm (for the investigated ECRISs 8.3 mT, assuming a 1 T radial flux density at the plasma chamber wall) seems to be small, nearly negligible. On the other hand, the plasma structure inside the plasma chamber clearly displays a hexapole distribution [8].

However, just assuming a hexapolar ion distribution at the extraction aperture is not satisfying [9]. Alternatively, the assumption of a streaming plasma inside the plasma chamber has a physical justification. Such streaming plasma is shown experimentally, for example, in [10]. There, a plasma formed by a vacuum arc flows into a magnetic solenoid exactly following the magnetic field lines. In that application, plasma is used to provide a target for a short-pulsed ion beam ($\mu$s) to provide a faster build-up time for the Space-Charge Compensation (SCC). In much larger machines also, such as tokamaks, the plasma is guided by the magnetic field lines. This general behavior is used to exploit the initial space coordinates of trajectories to be extracted from an ECRIS.

The superposition of the mirror field and the hexapole results in field lines with three loss zones on each side of the plasma chamber, see Figure 2, right, and in the center of the plasma chamber, where the radial component of the mirror field vanishes and six loss lines are obtained, same Figure, left. Only three of the six poles of the hexapole contribute to these lines where ions can be extracted; compare Figure 2 for experimental results and Figure 3 for the simulation result achieved with KOBRA3-INP [11]. Charged particles flow along these field lines with their specific Larmor radius. The ions are said to be "magnetized" when their Larmor radius is small compared to the characteristic system dimension, where the Larmor radius and the cyclotron frequency are given by:

$$r_L = m \cdot v_\perp / (q \cdot B) \tag{3}$$

and

$$\omega_c = (q \cdot B)/m \tag{4}$$

Some numbers for velocity (non-relativistic) and Larmor radius (also called gyroradius) for electrons and ions are given in Table 1. According to Equation (3), only the velocity component perpendicular to **B** contributes to the gyroradius, which is an unknown fraction of the assumed energy. The effective Larmor radius is therefore determined by a velocity which lies between 0 and the maximum velocity. The cyclotron frequency ($\omega_c$ in Equation (4)) is in this context only relevant to energetic electrons (those electrons accelerated to high energy by the microwave power). The table is for B = 1 T (yielding 28 GHz for $\omega_c$) and shows that not only are electrons magnetized but also low-energy ions. Even in the case where ions are not magnetized, they are guided by the magnetized electrons.

**Table 1.** Comparing Larmor radii with geometrical dimensions determines whether charged particles (electrons (e) and/or ions with m/q ratio of 1, 10, 100 atomic mass-to-charge units) are magnetized.

| $\frac{m}{q}$ | e | | 1 | | 10 | | 100 | |
|---|---|---|---|---|---|---|---|---|
| E[eV] | v[$\frac{m}{\mu s}$] | $r_L$[$\mu$m] | v[$\frac{m}{\mu s}$] | $r_L$[$\mu$m] | v[$\frac{m}{ms}$] | $r_L$[mm] | v[$\frac{m}{ms}$] | $r_L$[mm] |
| 1 | 5.9 | 0.4 | 0.14 | 150 | 4.4 | 0.46 | 14.0 | 1.4 |
| 10 | 19 | 11 | 0.44 | 460 | 14.0 | 1.5 | 44.0 | 4.8 |
| 100 | 59 | 34 | 1.4 | 1500 | 44.0 | 4.6 | 140.0 | 14.0 |

In Figure 3, the projection of some ion trajectories within the plasma chamber is shown. The low-energy ions gyrate along the magnetic field line and only those with adequate momentum can be extracted. Particles moving on a magnetic field line might be reflected by the magnetic mirror field, or if the initial velocity is high enough, they may enter the acceleration gap where ions will be accelerated and electrons decelerated and reflected. According to this model, the initial particle coordinates are generated in simulation only at those places where the attached magnetic field line passes through the extraction aperture without touching surrounding material. In an ECRIS, most of the magnetic field lines that pass through the extraction aperture originate in the plasma chamber in a primarily radial direction, producing material erosion and thermal load at these places. The sharp traces of erosion on the material indicate the small influence of diffusion. The reason for the typical ECRIS beam structure can be explained easily with this model.

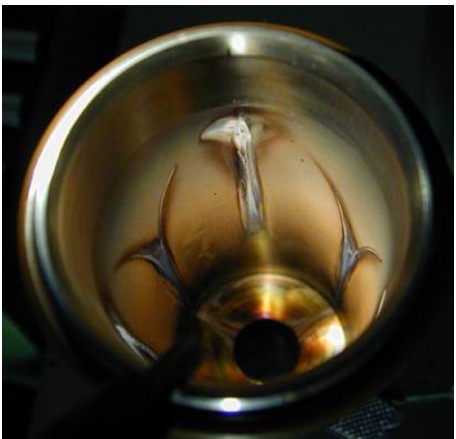 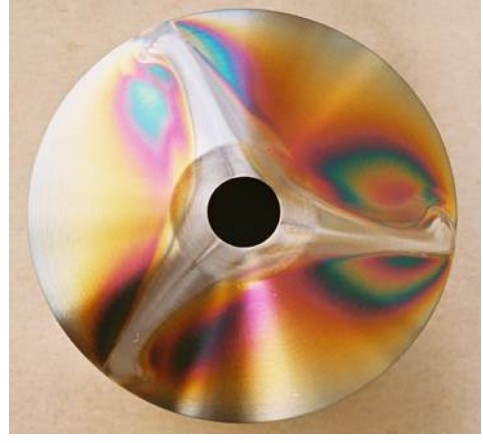

**Figure 2.** Left: six loss lines in the center of the plasma chamber and the hole for microwave injection. Right: the extraction electrode (plasma side) with three loss lines.

When the ions are accelerated within the extractor, they are still in the magnetic field of the ion source solenoid, where the Larmor radius translates to skewed trajectories. At the interface between plasma and extracted ion beam, the space-charge of the extracted ions will become dominant because the ion space-charge is not compensated by electrons any more.

Interestingly, note that the particle density will tend to homogenize along a magnetic field line inside the plasma chamber. The local density on each field line might be different from that on other field lines, resulting either in an emission-limited mode, or in a space-charge-limited mode at the plasma boundary, depending on the local **E**-field and the local ion density. Child's law [12] is still valid, but in comparison with the magnetic-field-free case, it is a local property now. As a consequence, the current density can vary across the extraction aperture.

Traces of erosion on the plasma side of the extraction electrode and on the plasma chamber wall contradict the assumption of a homogeneous plasma in front of the extraction aperture. Assuming that the motion of plasma is guided by magnetic field lines and thus that the plasma is collisionless, one concludes that only plasma formed on field lines that pass through the extraction aperture can be extracted. Close inspection of these field lines reveals that most of them hit the plasma chamber wall in a primarily radial direction, more strongly so for increasing radius at the extraction aperture.

Another dependency occurs: the longitudinal starting position of a field line depends on its azimuthal position. The closer the azimuthal distance within the plane of extraction is to a magnetic pole of the hexapole, the closer the longitudinal distance from this point on the field line to the extraction aperture. This might be a hint for the origin of the well-known structure shown by any ECRIS.

With increasing radius inside the plasma chamber, the azimuthal compression of plasma increases. Depending on the order of the multipole, this compression scales linearly with increasing r, the distance to the center, for the quadrupole, quadratic for the hexapole, and cubic for the octupole, and so on. This has to be taken into account for each trajectory. The initial momenta of the ions depend experimentally on the history of the creation process, and are still artificial data in simulation. Beside these spatial and velocity coordinates, the number of ions might be different for each trajectory, depending on the local plasma density. This radial and azimuthal dependency is characteristic for ECRISs.

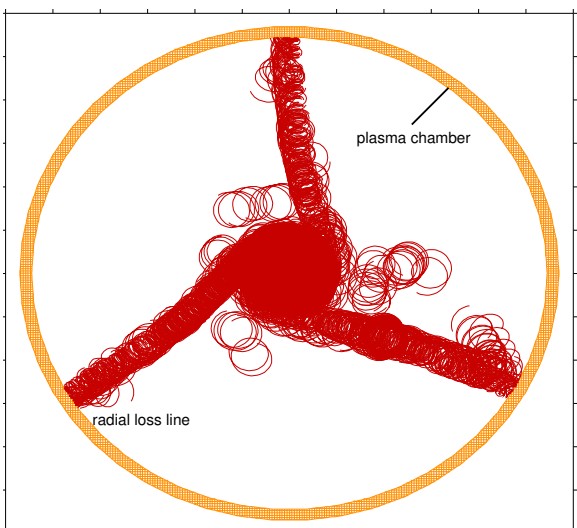

**Figure 3.** Ion trajectories (in red) within the plasma chamber (in orange), projected in the beam direction at 0.4 m with a projection depth of ±8 mm. The ions gyrate along the magnetic field line. See Figure 5 to identify this location close to the plane of extraction. The three wings in the plane of the extraction aperture have their origin in the radial plasma loss lines. In this projection, they appear like three $\rho$-shape wings, shifted by 120°.

In other kinds of ion sources, a three-grid extraction system (also called an "accel-decel" system) is commonly used, especially for higher particle densities, so as to maintain SCC of the extracted ion beam. Using a simple diode system leads to leakage of SCC. The negative potential drop formed by the middle ("suppressor") grid in a three-grid extraction system presents a barrier for low-energy electrons in the ion beam. Unsurprisingly, such systems are also required for ECRISs with high total extracted ion current. Experimentally, this influence can be seen by observation of the light coming from the plasma when switching on and off the negative potential dip. When the negative electrode is grounded, the plasma gets brighter due to the accelerated electrons from SCC.

For a correct simulation, the correct ion charge state spectrum must be used. This m/q spectrum can be measured or determined by simulation [13]. A set of differential balance equations describes the charge state distribution, which can be solved when cross sections for ionization, capture, charge exchange, and particle distributions are known.

Typically, the extraction electrode is located close to the maximum of the solenoidal field (i.e., the peak of the mirror field); the B-field decreases as the beam is accelerated within the extractor. Independently of the initial conditions, the different m/q-ratios are treated differently in the extraction system, due to the different velocities for each m/q. After acceleration of the ion beam, it simply drifts in a field-free region, or it may be still in the fringing field of the ion source magnets, or it may have already entered a focusing element. These different conditions translate into different beam transport properties.

## 3. Ion Sources

### 3.1. The CAPRICE Ion Beam Source

The ion source CAPRICE (see Figure 4 for the computer model) is a 14.5 GHz ECRIS version with a mirror field formed by two normal (non-superconducting) coils, providing 1.1 T for the first, and 1.0 T for the second coil maximum field on axis. The source has been operated with three different hexapoles, providing 0.8 T, 1.0 T, and 1.2 T flux density at the plasma chamber wall. The ion source has been described elsewhere in more detail [7]. Erosion marks of the plasma after some operation time inside the plasma chamber are shown in Figure 2.

Simulation of the magnetic field has been carried out using the OPERA [14] software and imported into the KOBRA3-INP software to perform ion beam simulation.

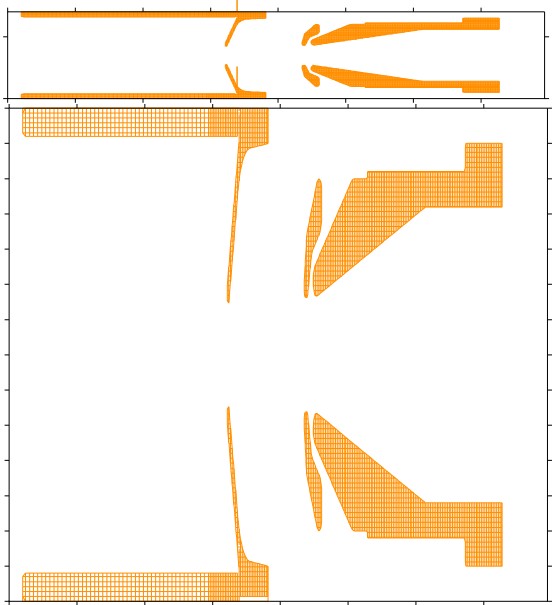

**KOBRA3-INP**
volume: 8×4×4 cm
nodes: 1,030,301
non-uniform mesh
potentials (l to r):
15, −2, 0 kV
plasma: 25 V
B-field: OPERA by
A. Andreev, GSI
CAD model by
R. Lang, GSI

**Figure 4.** CAPRICE computer model. Top: correct aspect ratio. Bottom: enlarged scale.

### 3.2. The HIISI, GHIISI Ion Beam Sources

The plan at GSI (Gesellschaft für Schwerionenforschung, Darmstadt, Germany) is to replace the 30-year-old CAPRICE ion source by a more modern type and the Finnish ion source HIISI [15] (see Figure 5 for the computer model) seems to be a favorable candidate. At the university of Jyväskyla (Finland), a normal conducting 18 GHz ECRIS has been built: Heavy Ion Ion Source Injector (HIISI). The HIISI ECRIS is an 18 GHz normal conducting version with improved magnetic properties compared to CAPRICE. To distinguish the Finnish version from the planned GSI version, the "G" stands for the planned version at GSI. Beside the higher frequency and slightly higher B-field, the computer model used is the same as used for CAPRICE. The magnetic field has been modeled using OPERA and CST [16], both of which show the same results, and can be used to calculate magnetic flux density tables and to import them to KOBRA3-INP. In general, both magnetic field distributions, for CAPRICE as for HIISI, are similar but for HIISI, higher flux densities are obtained. The cooling of the hexapole has been improved to allow higher losses from the plasma towards the plasma chamber without damaging the permanent magnets.

HIISI relies on an electrostatic lens system to focus the extracted ion beam to the following acceleration structure, whereas for GHIISI, it is planned to use the CAPRICE-type extraction system instead. Simulation should accompany this change to obtain best possible results, which is equivalent to the requirement to increase the brightness of the extracted ion beam. Brightness is defined as ion beam current per 4D-phase-space volume.

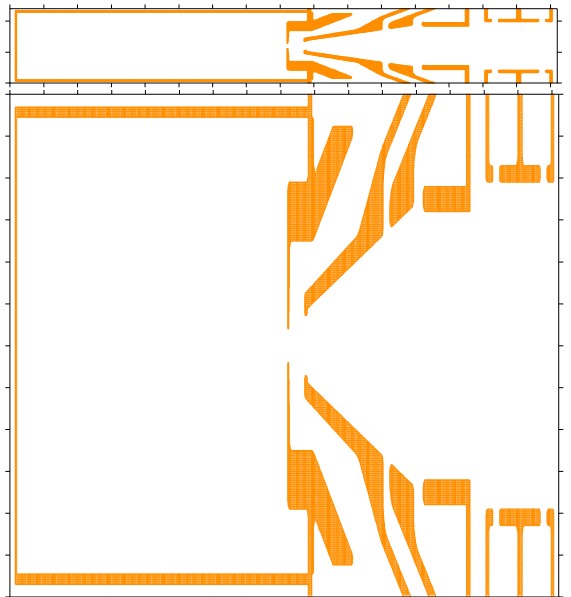

KOBRA3-INP
volume:
81×12×12 cm
nodes:     25, 171, 680
480 · 229 · 229
uniform mesh
potentials (l to r):
18, −3, 0, 10, 0, −10, 0 kV
plasma: 25 V
B-field:     CST     by
F. Maimone, GSI
CAD     model     by
R. Lang, GSI

**Figure 5.** HIISI computer model. Top: correct aspect ratio, bottom: enlarged scale.

## 4. Computer Experiments

### 4.1. Initial Coordinates of Trajectories

When using the "streaming plasma" model, the attached field line defines locations x, y, z where particles are generated. Magnetic field lines of interest are shown in Figure 6. Analyzing the path of magnetic field lines, it can be concluded that ions originated close to a radial loss line in the plasma chamber can be extracted as long as initial velocity components are sufficient. Using an adequate number of ion trajectories for each field line, the total number of trajectories for all field lines results in a very large (or even huge) number of trajectories to be calculated. To reduce the number of trajectories in the simulation, there are several options, for example, using not all places along a magnetic field line, but a representative selection of them only. For the magnetic flux density, a window (or a range) can be defined to create initial conditions only when B at this position is within that range.

A shell-like volume is defined with such a model. Each shell represents a resonance zone for a specific electron energy. Two different zones are the result: one zone when the magnetic field line enters the hollow shell, see Figure 7 and the other when leaving this shell, see Figure 8. Ray-tracing has been made for both cases and comparison of both results indicates that there is only a small influence of the hexapole on the ion beam profile when starting the ions close to the extraction aperture (Figure 7), whereas the other condition (Figure 8) shows a clear influence of the hexapole. The extracted ion beam (still no space-charge) is shown in Figure 9 by two cuts, one perpendicular to the beam direction, the other in the beam direction. After six iterations (solving Poisson's equation and doing ray-tracing, creating an updated space-charge distribution after each iteration), a self-consistent solution is found. The resulting potential including space-charge is shown in Figure 10.

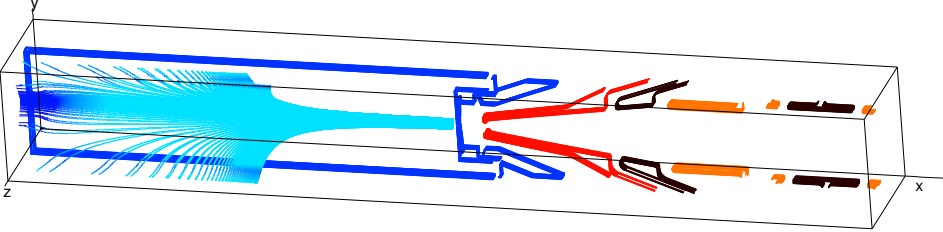

**Figure 6.** Projection of magnetic field lines going through the extraction aperture.

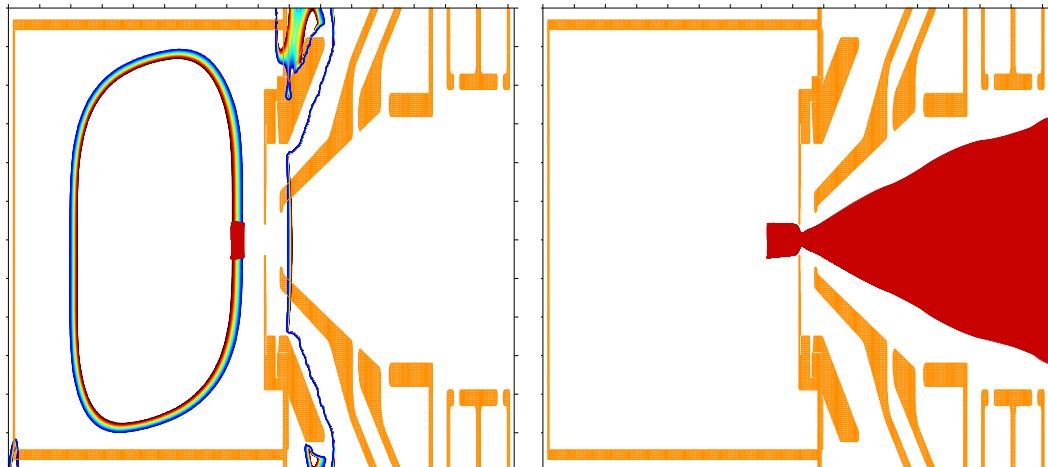

**Figure 7.** Initial conditions for ions created along a magnetic field line when B is between 1.0 T and 1.1 T. Left: lines of constant B for that range are shown here instead of field lines. Red points indicate the starting points for trajectories. Right: only the region of first intersection of constant B and a field line is used here. Assuming such a distribution of ions in front of the extraction system, a nice beam can be simulated, but it does not represent the reality.

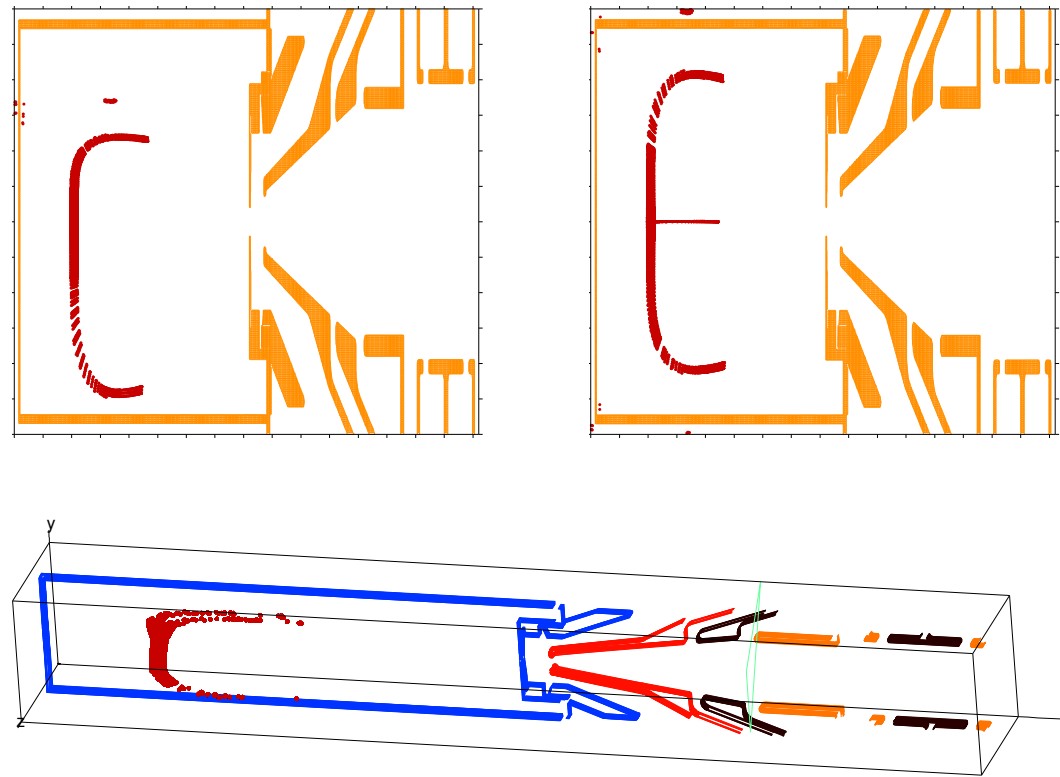

**Figure 8.** Ions are started in the same B region as in Figure 7, but here, the second intersection of constant B and a field line is used. Top left: vertical projection. Top right: horizontal projection. Bottom: 3D projection. In this figure, only the initial coordinates of trajectories are shown.

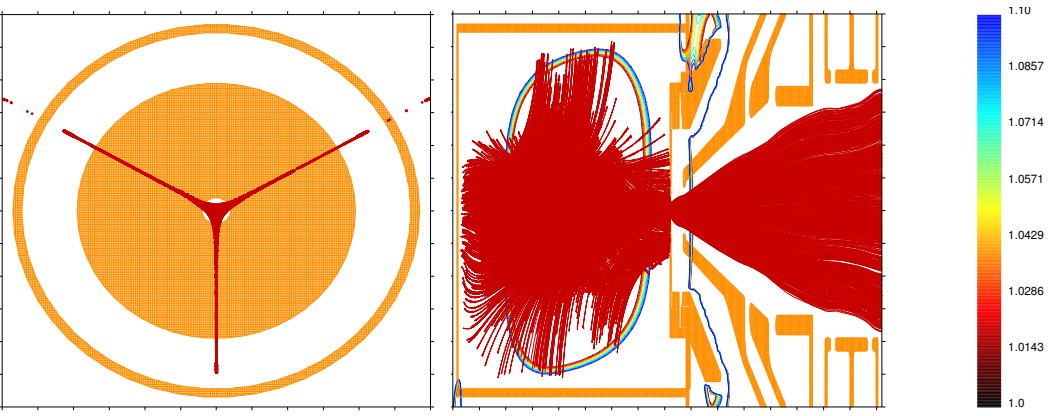

**Figure 9.** Left: projection of initial positions of trajectories in longitudinal direction at x = 0.4 m. Right: trajectories in a transverse projection, at 6 cm with ±6 cm projection depth. Lines of constant B from 1.0 T to 1.1 T are shown together with the trajectories.

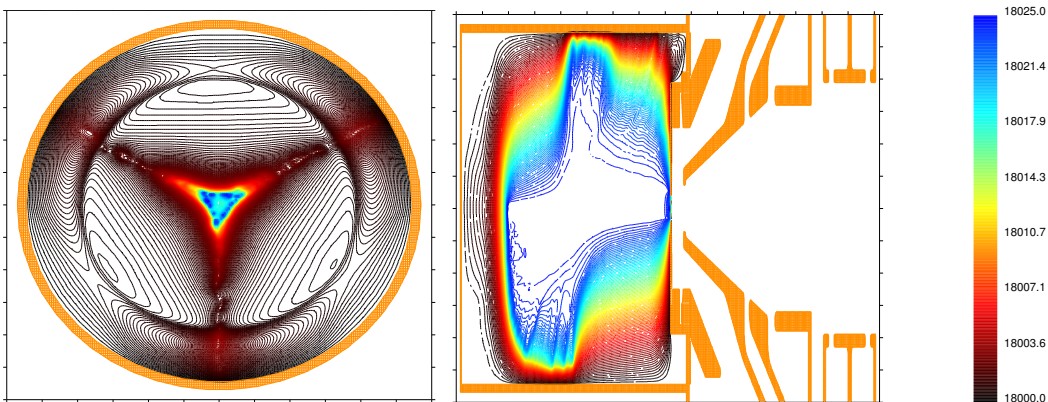

**Figure 10.** Potential distribution within the plasma chamber. Only three "active" loss lines are used to describe the plasma. Lines of constant potential from extraction electrode potential to plasma potential are shown. Left: cut in the beam direction just in front of the extraction aperture. The distorted azimuthal potential distribution is caused by space-charge. Right: cut in the transverse direction (perpendicular to beam direction).

### 4.2. Current Density Distribution

The model does not provide the initial current density for each trajectory, but a perveance scan will show at least the tendency for the extracted ion beam current under the influence of space-charge. In distinction to the magnetic-field-free case, the homogeneity of the ion beam is defined by the number of charged particles available on each **B**-field line.

The plasma boundary cannot be visualized experimentally, but simulation can be used to give a picture of this theoretical description. The potential distribution at the extraction aperture does not have any azimuthal dependency caused by geometry. Due to the particle distribution within the plasma chamber and in front of the extraction aperture, the potential distribution at that place shows the influence of the hexapole. In Figure 10, the potential distribution shows a triangular shape caused by space-charge of the incompletely compensated plasma (here plasma potential is 25 V, electron temperature 5 eV). Choosing such numbers neglects the space-charge of accelerated high-energy electrons. Only cold electrons will contribute to the plasma potential. More exact parameters for that would be available if the energy distribution of electrons would be known. In the longitudinal cut, it can be seen that the plasma chamber is not fully filled with plasma (potential), there are empty spaces, and the potential there is lower than the plasma potential. The distorted azimuthal potential distribution within the extraction plane might increase the beam emittance (any nonlinear force will increase the emittance).

### 4.3. Frequency Tuning, Multiple Frequencies

Tuning the rf-frequency by a small amount (MHz steps in the GHz range) shows a strong influence on the extracted ion beam current [17]. This influence on the total extracted ion current can only be explained by electrons which are influenced in their spatial and energy distributions. Ions are not sensitive to the GHz frequency of the rf-power.

It sounds plausible that tuning the frequency affects the plasma density on each B surface which is connected to a specific electron energy. In the simulation, this can be approximated by choosing a certain range of B, representing such a shell described by Equation (4). The same argument holds for several simultaneously injected frequencies for heating the electrons. Further experimental work needs to be done to verify or to discard the model in that aspect.

### 4.4. Biased Electrode

It has been found experimentally that an additional electron emitter has a strong influence on the ion beam intensity [18]. Such an emitter is located on the opposite side of the extractor. Electrons coming from such a biased disc see comparable magnetic fields as

experienced by ions on the extraction side. They will not stay on axis, but drift to radial loss lines instead, see Figure 11. From this, it should be concluded that electrons coming from a biased disc are mainly accelerated to higher energy and do not contribute to SCC of plasma. The space-charge of, e.g., 200 eV electrons (for the voltage drop of a disk biased to −200 V) is much smaller than that from a few eV energy distribution of electrons.

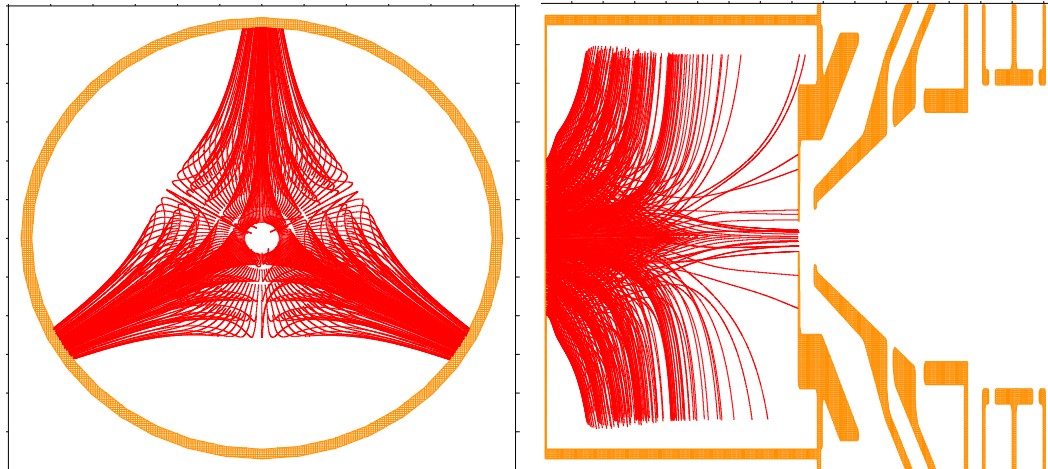

**Figure 11.** Projection of electron trajectories, started at the disc, biased to −200 V. Left: cut perpendicular to the beam direction within the plasma chamber. Right: cut in the beam direction. The structure is caused by discretization in radius and azimuth of the electron "beam".

From that simulation, a biased ring should be applicable as well, which would solve the problem of blocking this position by another element (microwave entrance port, oven, etc). Maybe even more interesting could be a biased ring on the extraction side because the three "emitting" poles would be provided with additional electrons.

### 4.5. Afterglow

For pulsed operation (as, for example, for synchrotron injection), the afterglow effect can be used to enhance the available ion intensity. It has been found that when switching off the rf-power, an ion beam can still be extracted, with even higher intensity compared to the intensity when the rf-power is on. This afterglow effect can be explained by the capacity of each **B**-field line to store charged particles. If rf-heating is stopped, the "capacitor" will be discharged. To estimate this capacity, this length of the magnetic field line (from the plasma chamber wall to the extraction aperture) could be used. The larger in diameter and longer in length, the stronger the afterglow effect should be according to the model. However, the maximum number of stored ions along the **B**-field line depends also on the number of electrons frozen to that field line. The higher this number, the higher the capacity. Combining the afterglow effect with a biased ring in front of the extraction electrode should improve yet further the available intensity in that operating mode.

### 4.6. Other Effects

It should be mentioned that there are experimental observations which cannot be explained by the proposed model, such as the so-called isotope effect [19], indicating that the model is still not complete.

### 5. Transport Issues
#### 5.1. Beam Properties

After extraction, the ion beam properties can be best described by its emittance. This quantity is a projection of the 6D-phase-space onto a 2D drawing plane. The six dimensions are the three spatial coordinates and three momenta. Typically, the three momenta are normalized to the longitudinal momentum, resulting in an angle for both

transverse directions. The different projections at a given longitudinal position, integrated over all longitudinal velocities, are the profile in space, the profile in momentum space, two emittances (horizontal, vertical) and two remaining projections, called mixed phase-spaces. They are shown in Figure 12. The typical shapes from an ECRIS are reproduced by simulation with KOBRA3-INP. Different charge states of Argon ($Ar^+$, $Ar^{2+}$ and $Ar^{3+}$) are distinguished by color in this figure. The expected different azimuthal rotation for different m/q values is confirmed by simulation.

The profile of the ion beam shows a triangular shape. More precisely, the extracted ion trajectories are better described by three wings separated from each other by 120°, more distinctly so with increasing radius; see Figure 12, top row, left column. Both emittances at that position show a divergent beam (top row, center and right). Some regions of more dense distribution are caused by the projection of this triangular shape. However, the projection of the 6D-phase-space in other directions unveils that the initial structure stays embedded, see same Figure, bottom row: transverse angle against horizontal position and horizontal angle against vertical position as well as the momentum space (horizontal angle against vertical angle).

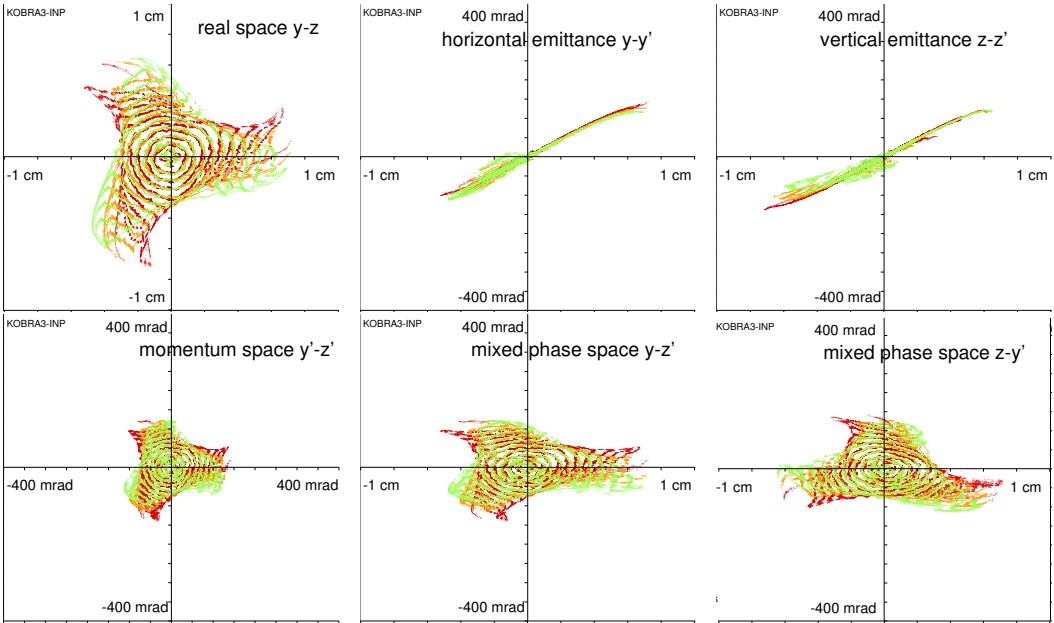

**Figure 12.** Different projections of the 6D-phase-space behind the extraction electrode (x = 0.45 m in Figure 5). Top row: real space, horizontal, and vertical emittance. Bottom row: momentum space, horizontal, and vertical mixed phase-spaces (horizontal angle against vertical position and vice versa). Scaling ±1 cm, ±400 mrad. Green: $Ar^+$, brown $Ar^{2+}$, and red $Ar^{3+}$. Each of these six projections contains 465,688 trajectories.

Let us assume that the ion beam application is as a particle injector to an accelerator, as quite usually the case. The aim of optimization is therefore to achieve a beam with highest possible intensity and lowest possible emittance of the desired m/q ratio ion. The correlation which is shown in Figure 12 needs to be taken into account for the design of the beam line for an ECRIS beam [20]. To measure these different projections, a pepper pot emittance meter such as the one designed by H.R. Krämer [21] can be used, or the $\mu$-Faraday cup array built by L. Panitzsch [22].

As soon as the ion beam is in a field-free region after extraction, the ions proceed on straight trajectories without any change in angle. This is no longer true when the ion beam current density is high. Then, the space-charge of the extracted ion beam becomes important. The potential drop $\Delta\Phi$ across a homogeneous parallel beam can be estimated as $\Delta\Phi = 30I/\beta$ ($I$ ion beam current in [A], $\beta$ ion velocity divided by the speed of light). The resulting force acts as a de-focusing lens. However, ions extracted from an ECRIS, as

for any ion source, might collide with residual gas atoms inside the beam line, ionizing them with a certain probability. The newly created ions are repelled by the positive space-charge potential of the extracted ion beam but electrons are trapped by this potential. Electrons with energy higher than the remaining potential hill of the positive ions will escape; therefore, only the coldest electrons will remain trapped within the beam potential. Depending on the creation rate and loss rate of those electrons, a certain time is required to reach space-charge neutrality. When operating the ion beam continuously, it is just a question of time until space-charge neutrality is reached; for pulsed operation, the process repeats for each pulse. Typical build-up times are in the range of $\mu$s to ms, depending on several parameters. However, when ion-optical elements are used to transport the ion beam inside the beam line, SCC will be influenced by these elements as well.

The magnetic self-field scales with $\beta$ which can be neglected for ion beam extraction. This is not valid for electrons above several keV due to the higher velocity and typically higher currents compared to ions.

### 5.2. Drift, Electrostatic Plasma

The ion beam drifts inside the vacuum of the beam-line without any external field. Electrons for SCC can move freely within the ion beam. Only the electric field caused by the space-charge of the extracted ion beam influences the SCC. An electrostatic plasma will develop, formed by fast moving ions and low energy electrons. Movement of electrons is governed by the electric field caused by the space-charge of ions. The lower the electron energy, the lower the space-charge potential that develops with time. The low-energy electrons within the Coulomb cloud of the positive ion beam can move freely and can be described as an electrostatic plasma.

### 5.3. Ion Beam Optics with Electrostatic Elements

Different electric devices are available to bend or focus the ion beam, such as Einzel lenses, dipoles, quadrupoles, and more. Their common feature is that the SCC is influenced by them as well—the electric field produced by the specific element will influence the SCC. It will tend to counteract the field produced by the ion beam's space-charge. This acts as a nonlinear lens creating image errors and emittance growth. Negatively biased electrodes repel electrons, leaving a part of the ion beam uncompensated, whereas a positively biased disc will remove electrons from SCC of the ion beam. With increasing space-charge, such lenses become impractical. Higher current capability of power supplies for positively biased electrodes as well as improvement of cooling of these electrodes will be required. Finally, SCC will suffer due to these electric fields.

### 5.4. Ion Beam Optics with Magnetic Elements

In using a magnetic element capable of focusing or bending the extracted ion beam, it is evident that low energy electrons of SCC are magnetized by the magnetic field. They can move freely in the direction of the magnetic field, but are frozen in the direction transverse to the magnetic field.

It has been shown experimentally that SCC of an ion beam can be maintained even within transverse magnetic fields as in a magnetic dipole used to separate ions of different m/q. In the second half of the 1980s, the author had the pleasure of meeting Boris Makov at an ion source conference at Dubna. Makov was responsible in the 1940s for enrichment of $^{235}U^+$ by ionizing natural uranium in a PIG ion source and separating the different isotopes within the ion beam by a magnetic dipole. The author's question about the SCC of the ion beam including the bending magnet section was answered with a "knowing smile" on his face with: "Otherwise the transport and separation of more than 100 mA of uranium with 30 keV energy would not have been possible". Experience. In [1], this experience was confirmed by 3-dimensional simulation some years later. Electrons oscillate along the magnetic field line driven by the electric field caused by space-charge of the ion beam. SCC can build up.

It has to be concluded that electrostatic lenses will destroy SCC, at least partially, whereas transport with magnetic lenses maintains SCC for ion beam transport after the required build-up time. A consequence for pulsed operation is that the build-up time adds up along the beam line and thus shortens the effective pulse length. The local SCC for cw operation is a property of the beam line. Similarities between the plasma behavior within the plasma chamber and the beam plasma are not accidental.

## 6. Conclusions

Simulation of a system without exact knowledge about the appropriate model is not useful. In addition, a reliable model needs to be compared with experimental facts. Without compliance, no confidence to simulation results can be achieved. The simulation results obtained with KOBRA3-INP using a model specific to ECRIS show good agreement with experimental experiences with that type of ion source. It shows how the volume of the 6D phase-space of the ion beam is treated by the combination of electric and magnetic fields. The hexapolar component of the magnetic field and its influence on the extracted ion beam is responsible for the well-known structure obtained by any ECRIS. Beside the m/q distribution, the extracted ion beam is composed of three spatially-separated fractions coming from three poles of the hexapole. Empty spaces in the 6D phase-space are causal for the system and are not removable by Hamiltonian forces.

Use of this model allows, for example, a comparison of the radial confinement characteristics by different multipoles by simulation, at least from the standpoint of ion beam extraction; how the plasma-generator is affected by different multipoles is not (yet) described in that model. The list of possible parameter studies can be extended easily: magnetic shape, additional electron emitter, different frequencies, etc.

The model is also applicable for SCC simulation and understanding of further ion beam transport of the accelerated ion beam along the beam line, equipped with different types of beam-optical elements.

**Funding:** This research received no external funding.

**Institutional Review Board Statement:** Not applicable.

**Informed Consent Statement:** Not applicable.

**Data Availability Statement:** Not applicable.

**Conflicts of Interest:** The authors declare no conflict of interest.

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
