# Peer review of "Simulation as a Tool for Understanding Experimental Observations—Ion Beam Extraction from an ECRIS"

_plasma, doi:10.3390/plasma5040038_

Round 1
Reviewer 1 Report
The paper is a summary and review of the author's work on the computer simulation of ion beam formation from electron cyclotron resonance ion sources (ECRISs), and includes some new and novel thinking (and associated simulation results) about the significance, in ECRISs, of ion motion being strongly tied to magnetic field lines in collisionless plasma. This insight provides good explanation for the plasma-etched loss-lines seen nearly universally on the metal plasma chamber wall of ECRISs. The layout of the paper is logical and clear. The English is excellent. The paper will make an important contribution to the ECRIS literature and will be of great interest to workers in this field.
Author Response
Dear reviewer,
thank you very much for your referee. Nothing to be changed.
With my very best regards
Peter Spädtke
Reviewer 2 Report
Author described the model for the simulation of the beam extraction from ECRISs. It is important and useful for investigating the quality and trajectory of the extracted beam form ECRIS
To understand it more clearly, I give several comments as follows
1)The author put Figure 6 in page 8. However, there is no description in the text. If it is not important, I suggest that the author omits this figure from the article.
2)The author shows the line of constant B in figure 7. The line of constant B is strongly dependent on the magnetic field distribution. I suggest that the author shows the magnetic field strength (Binj, Bmin, Bext and Br)
3)P10L252. "In figure 10 the potential distribution shows a triangular shape caused by the space-charge of the incompletely compensated plasma (here plasma potential is 25 V, electron temperature 5 eV)"
Usually, the plasma potential is dependent on the ion source conditions (RF power, gas pressure, magnetic field distributions etc). I suggest that the author describes the ion source conditions to obtain the plasma potential of 25 V.
Author Response
Dear reviewer,
thank you very much for your referee. I would like to answer your three comments.
1) I did reference Figure 6. It just appears in the text after the Figure (line 228, page 8). I think that this picture is important because it shows where the magnetic field lines which are going through the extraction aperture are coming from.
2) I agree that the description of the magnetic flux density using Bz(z), Br(r), Bext and Bmin are useful quantities to describe the nagnetic field configuration. However, in this context described here the additional information provided by the trajectory of the magnetic field line is important. I will add a sentence for that in the text.
3) I agree that I did not mention how to achieve plasma potential and electron temperature in experiment. In my simulation I refer to plausible numbers as sometimes mentioned in literature. I am aware that the definition of temperature with their given energy distribution of electrons is problematic.
I will add a sentence to that in the text.
With my best regards
Peter Spädtke